# The Role of Tryptophan Metabolites in Neuropsychiatric Disorders

**DOI:** 10.3390/ijms23179968

**Published:** 2022-09-01

**Authors:** Majid Davidson, Niloufar Rashidi, Kulmira Nurgali, Vasso Apostolopoulos

**Affiliations:** 1Institute for Health and Sport, Victoria University, Melbourne, VIC 3011, Australia; 2Regenerative Medicine and Stem Cells Program, Australian Institute of Musculoskeletal Science (AIMSS), Melbourne, VIC 3021, Australia; 3Department of Medicine Western Health, Faculty of Medicine, Dentistry and Health Sciences, University of Melbourne, Melbourne, VIC 3010, Australia; 4Immunology Program, Australian Institute of Musculoskeletal Science (AIMSS), Melbourne, VIC 3021, Australia

**Keywords:** tryptophan, neuropsychiatric disorders, metabolic pathways, pharmacological influences

## Abstract

In recent decades, neuropsychiatric disorders such as major depressive disorder, schizophrenia, bipolar, etc., have become a global health concern, causing various detrimental influences on patients. Tryptophan is an important amino acid that plays an indisputable role in several physiological processes, including neuronal function and immunity. Tryptophan’s metabolism process in the human body occurs using different pathways, including the kynurenine and serotonin pathways. Furthermore, other biologically active components, such as serotonin, melatonin, and niacin, are by-products of Tryptophan pathways. Current evidence suggests that a functional imbalance in the synthesis of Tryptophan metabolites causes the appearance of pathophysiologic mechanisms that leads to various neuropsychiatric diseases. This review summarizes the pharmacological influences of tryptophan and its metabolites on the development of neuropsychiatric disorders. In addition, tryptophan and its metabolites quantification following the neurotransmitters precursor are highlighted. Eventually, the efficiency of various biomarkers such as inflammatory, protein, electrophysiological, genetic, and proteomic biomarkers in the diagnosis/treatment of neuropsychiatric disorders was discussed to understand the biomarker application in the detection/treatment of various diseases.

## 1. Introduction

Psychiatric disorders (PDs) (i.e., bipolar/mood disorders, schizophrenia) are prevalent ailments that have a devastating effect on the health and well-being of sufferers. In the previous century, medical-based information regarding the neurobiology of PDs was deficient. However, many uncertainties have been elucidated over the last few decades, and new horizons toward developing efficient treatments for mental illnesses have been opened. Recent studies suggest that the kynurenine pathway (KP) as a biological pathway has excessive potential for recognizing major psychiatric disorders [1,2]. Tryptophan (TRP) is the most prevalent amino acid, which plays a significant role in protein biosynthesis in humans/animal bodies. Apart from its incorporation into protein, TRP is metabolized according to disparate routes and produces biologically important components [3,4]. The synthesis of TRP follows two metabolic pathways [5,6,7]:

Kynurenine pathway (KP) creates nicotinic acid as a precursor to nicotinamide adenine dinucleotide coenzymes.Serotonin pathway (SP) eventuates in the production of serotonin as a neurotransmitter and melatonin as a neuromodulator.

Previous studies have proved that TRP and some of its metabolites, such as serotonin, melatonin, and 5-hydroxytryptophan, can be included in melanogenesis due to their ability to be metabolized by applying tyrosinase and peroxidase [8,9,10]. TRP and its metabolites play a substantial role in alleviating disparate types of illnesses ranging from psychiatric/neurological diseases to cancer. This brilliant characteristic has made the investigation of TRP metabolism an exciting field of study for biomedical researchers focusing on developing novel therapeutic target identification [11,12,13]. Recent research illustrates that the KP can improve numerous biological systems that act inefficiently in psychiatric disorders like central nervous system (CNS) neurotransmission and immune-inflammatory systems [1,14]. Appropriate interpretation of the linkage mechanisms of the kynurenines (KYNs) with various systems is an excessive interest [15]. Although disrupted KP enhances the risk of PDs, suitable treatment approaches such as electroconvulsive therapy, physical exercise, and non-steroidal anti-inflammatories change the metabolism of KYN [16,17]. As such, these features can result in evaluating the potential of KYN metabolites as promising therapeutic biomarkers. Moreover, the targeting process of the KP eventuating is an outstanding opportunity to develop effective treatments for neuropsychiatric disorders [18,19].

A biological marker (biomarker) can be identified as an objective indicator of biological, pathological, or pharmacological response to medical-based interventions [20]. Some physiological biomarkers are also known as indicators of the body’s physiological functioning, including heart/breathing rate [21]. Biomarkers possess great potential for application in assessing drug impacts, diagnosis, and monitoring of clinical response in patients suffering from neuropsychiatric disorders [20].

Herein, we open new horizons towards a better understanding of the impact of TRP metabolites on neuropsychiatric disorders by discussing the existing pathways of its metabolism. Additionally, TRP and its metabolites quantification following neurotransmitter precursors are included, which represents the novelty of this work. A comprehensive evaluation of the efficacy of the different biomarkers, including inflammatory, protein, electrophysiological, genetic, and proteomic biomarkers in detecting neuropsychiatric disorders, is also presented and provides an appropriate opportunity to understand the potential applications of biomarkers.

## 2. TRP Metabolism

TRP is a vital part of the human diet and plays an essential role in neuropsychiatric health, physiological stress responses, inflammatory responses, oxidative systems, and gastrointestinal (GI) health [22]. High tryptophan foods include turkey, chicken, pork, red meat, fish, milk, beans, nuts, eggs, and oatmeal. The prominent catabolic pathway for TRP is the KYN, which is equal to almost 95% of dietary TRP degradation [6,11,12]. The importance of this pathway is significant for the biosynthesis of niacin. Sixty milligrams of TRP results in the production of 1 mg of niacin [23]. Conversion of the majority of dietary TRP to non-aromatic compounds is often performed using the glutarate pathway. The catalysis process of the first phase of TRP degradation using TRP 2,3-dioxygenase (TDO) and indoleamine 2,3-dioxygenase (IDO-1 and IDO-2) results in the formation of N-formyl kynurenine that is quickly converted to KYN by N-formylkynurenine formamidase. Although the enzymes mentioned above catalyze similar oxidation reactions to form N-formylkynurenine, their mechanical operation is different. The responsibility of TDO is to metabolize L-TRP by applying molecular oxygen as a cofactor [24,25].

In comparison with TDO, IDO has inclusive substrate specificity and can operate on other indole derivatives. They apply superoxide anion and molecular oxygen for activity [26,27]. IDO can be present in two forms as follows [28].

IDO-1: this form of IDO exists in most non-hepatic tissues.

IDO-2: this form of IDO exists in the kidney, liver, and brain.

IDO-1 is a noteworthy physiological regulator of immune system activation, expressed in dendritic cells, secondary lymphoid organs, and pancreatic β-cells [29,30]. IDO-1 expression possesses a particular role in providing peripheral tolerance in autoimmune diseases and cancer. It is worth mentioning that IDO-1 and IDO-2 enzymes can play an essential role in the catabolism of L-/D-tryptophan and their metabolites, including 5-hydroxytryptophan, serotonin, melatonin, and 5-hydroxyindolacetic acid [31]. Therefore, it has been propounded that IDO-2 can be applied as a negative IDO-1 regulator via a competitive mechanism. The regulation process of IDO enzymes can be done by interferon γ as a pro-inflammatory cytokine, bacteria lipopolysaccharides, viruses, and tumor cells [32,33,34].

## 3. Neurotransmitters Precursor

The synthesis process of serotonin (an important neurotransmitter) is performed using TRP via the 5-hydroxytryptophan (5-HT) pathway. In some organs, such as the intestine, conversion of serotonin into N-acetylserotonin takes place employing serotonin-N-acetyltransferase (SNA). Then, SNA is converted to melatonin during the hydroxyindole-O-methyltransferase action. Serotonin is an essential chemical neurotransmitter, and its presence is of great importance in regulating mood, sleep, and appetite [34,35]. Melatonin is considered a vital neuromodulator, which has shown great potential as a strong antioxidant to regulate numerous body-related activities such as circadian rhythm and pain [36,37,38,39]. Of note, only free TRP fractions possess the capability to move through the blood–brain barrier to reach the brain [5,40].

Three parameters, including the concentration of free TRP, the concentration of neutral amino acids, and meal carbohydrates (that conduct some neutral amino acids from plasma to muscle), are essential to identify the amount of TRP converted to serotonin in the brain. The occurrence of functional imbalances in the formation of downstream metabolites during the activation process of the KYN pathway is believed to be the prominent reason for disparate central/peripheral sicknesses such as depression, aggression, schizophrenia, and acquired immunodeficiency syndrome [2,41,42]. Among KYN metabolites, quinolinic acid (QA) and kynurenic acid (KYNA) attract attention due to their capability to bind the N-methyl-D-aspartate (NMDA) receptor and their potential impact on developing neuropsychiatric diseases [43,44]. QA boosts the excitotoxic neuronal damage due to its activity as an agonist for the NMDA receptors. In contrast, KYNA has shown great neuroprotection potential due to operating as an NMDA antagonist.

Moreover, KYNA can act as an inhibitor of the α7 nicotinic acetylcholine (α7nACh) receptors [45]. It is essential to have a balance between the values of neurotoxic QA and neuroprotective KYNA under physiological circumstances. As such, the emergence of various pathologic conditions is mainly due to the imbalance in the generation of QA and KYNA (higher values of QA), which enhances neurotoxicity. For instance, in neuroinflammatory diseases, the release of pro-inflammatory cytokines eventuates to improve the amount of QA and decrease KYNA by increasing TRP degradation via the KYN pathway [46,47,48].

## 4. TRP and Its Metabolites Quantification

Due to the indisputable relationship between TRP metabolism and pathophysiology, modifications/alterations in TRP and its metabolites in disparate biological fluids such as plasma, serum, and urine have been studied to perceive potential biomarkers [3,49,50]. Conventional techniques for analyzing TRP and its metabolites are liquid/gas chromatography connected with different detection systems such as UV and fluorescence/mass spectrometry [51,52,53,54]. The advancement of high-technology procedures has considerably revolutionized the analysis of TRP and its metabolites. For instance, Sadok et al. developed the ultra-high-performance liquid chromatography-electrospray ionization-tandem mass spectrometry (UHPLC-ESI-MS/MS) technique to quantify TRP and KP metabolites biologically and validated it independently for both matrices used in serum and peritoneal fluid analysis. They concluded that this model is excellent for evaluating TRP and KYN in blood and peritoneal fluid [55].

## 5. The Role of TRP in Psychiatric Disorders

Alteration in the value of circulating TRP can result in a significant inconsistency in its accessibility for 5-HT/ melatonin (MLT) synthesis in the brain. Thus, it may be involved in the pathophysiology of disparate neuropsychiatric disorders. Indeed, the imbalanced process of 5-HT neurotransmission can be attributed to the pathophysiology and neuro-psychopharmacology of various psychiatric diseases [32,36,37,56,57]. Numerous studies have evaluated serum and cerebrospinal fluid (CSF) TRP as well as other psychoactive formed components by employing the KYN pathway. Despite several investigations about the serum and CSF TRP in various pathologic circumstances, only very few evaluated the correlation between serum and CSF TRP concentration [58,59,60]. A declinine of CSF and TRP has been linked to different types of psychiatric disorders and clinical symptoms. It is worth pointing out that with the existence of comorbidities such as suicidal behavior, depressed patients illustrate higher decrement in circulating TRP compared to those without these comorbidities [61,62,63,64]. Surprisingly, the simultaneous presence of two diseases such as obsessive-compulsive disorder (OCD) and attention deficit hyperactivity disorder (ADHD) symptoms may increase the amount of TRP [65,66,67,68,69]. Ramos-Cha’vez et al. illustrated that anxiety positively correlates with depression scores and negatively correlates with TRP in a cohort study including 77 women over 50 [70]. Eventually, the simultaneous occurrence of depression and anxiety (in almost 50% of patients) and their symptoms may be similar. Recent evidence has corroborated the relationship between TRP and inflammation on anxiety [3,71]. The significant variation in the KYN pathway in the CNS has emerged to induce serotonin/melatonin deficiency.

Additionally, increased TRP catabolites (TRYCATs) can retain anxiety due to performing as endogenous anxiogenic [72,73]. However, those scientific investigations which evaluated peripheral TRP in depression did not consider comorbidities known to change TRP. This restriction is a major approach that needs to be considered. Although most reports show reduced plasma or serum TRP in depression, other reports contradict these findings [74]. Table 1 enlists disparate employed biomarkers for neuropsychiatric diseases.

Bipolar disorder (BD) is a mental health condition that causes extreme mood swings that include emotional highs (mania or hypomania) and lows (depression) [75]. Those patients suffering from this chronic disorder may be followed with a higher prevalence of obesity than the general population, which eventuates the risk of metabolic and cardiovascular sicknesses and, consequently, lower life expectancy [76,77,78]. Moreover, obesity as a chronic disorder is accompanied by cognitive deficits, a more significant number of suicide attempts, and shorter periods of euthymia [79,80]. From a historical aspect point of view, diagnosis and treatment of BD are only based on clinical phenomenology. Perception of efficacious biomarkers with the capability to prepare evidence about the current stage of disorders and estimate the further course of the illness is an essential effort in BD scientific investigation. TRP is of great importance in the biological underpinnings of BD through its catabolic pathways [81]. TRYCATs are involved in the pathophysiology of mood disorders by mediating immune inflammation and neurodegenerative processes. A meta-analysis of 21 eligible studies showed that the TRYCATs pathway is downregulated in BD patients. However, further studies are required to confirm the association of peripheral and central TRYCATs levels and BD [82]. TRP may be converted to 5-HT, serotonin, N-acetylserotonin, and melatonin. The Kynurenine-axis is another metabolic pathway that relies significantly on the enzymes indoleamine IDO-1 and TDO [83]. Despite acceptable stability in the activity of TDO, the activity of IDO-1 is greatly enforced in monocyte-derived cells via pro-inflammatory cytokines such as IFN-γ, IL-2, and -6 or TNF-α [84].

Schizophrenia is a neuropsychiatric disorder related to the human body’s TRP level. For the first time, the reduced plasma concentration of TRP was recognized in patients with schizophrenia aged over 40 years old [41,85,86]. Due to the metabolism of more than 90% of TRP via the kynurenine pathway controlled by rate-limiting enzymes IDO and TDO, the decrease of TRP in schizophrenia may be attributed to an enhanced conversion of TRP to kynurenine metabolites [87,88]. IDO regulation of KP metabolism is of great importance in modifying myelin-specific T cells’ activation [86,89]. These activated T cells have shown brilliant performance in creating pro-inflammatory cytokines, which directly contribute to demyelination and indirectly strengthen antibodies against myelin proteins [90,91,92]. In fact, a collection of evidence has illustrated that white matter abnormality is an essential factor of schizophrenia pathophysiology and may contribute to symptoms of schizophrenia [93,94].

Major depressive disorder (MDD) is known as the prominent reason for disability worldwide and thus can be regarded as a public health concern [95]. However, recognizing the pathophysiology of this critical psychiatric disorder is still a big challenge. In current decades, serotonin has been extensively associated with major depression with the classical assumption of low serotonin levels in the CNS [96]. Evaluation of peripheral serotonin in a major depressive episode can be an essential part of the investigation due to the storage of serotonin in the periphery [97]. Numerous available papers have investigated peripheral serotonin levels in blood, platelet, and plasma and demonstrated lower concentrations in MDD sufferers compared to control arms [98,99,100,101]. TRP is identified as the precursor of serotonin, which can be converted to 5 hydroxy-TRP and then serotonin. However, the decrement of peripheral levels in the major depressive episode is unclear [102,103,104]. Thus, the observed reduction in peripheral serotonin levels in major depressive episode patients may result from a nominal synthesis rate or a high turnover. Peripherally, the synthesis process of serotonin relies on the accessibility of TRP and two enzymatic conversions [105]. To assess the high turnover rate, an increase in metabolite levels must be seen. Investigation of these assumptions needs a complete evaluation of the serotonin pathway like serotonin, its precursors, and its metabolite in major depressive episode patients in comparison with controls arms. It has been corroborated that the lower levels of kynurenic acid can be accompanied by severe cognitive problems in major depressive episode patients [106]. Moreover, former investigations have demonstrated that xanthurenic acid possesses a great ability to modulate synaptic transmission in the hippocampus (a highly contributed structure in major depressive episodes [107]).

Attention deficit hyperactivity disorder (ADHD) is one of the most common behavioral disorders, characterized by impulsivity, hyperactivity, and inability to have attention. Although ADHD has high prevalence and causes drastic functional impairment, the neural basis of the condition still remains poorly understood [108]. It was initially hypothesized that decreased 5-HT could result in heightened impulsivity. However, further studies showed that the correlation between TRP metabolites and ADHD is more complicated, possibly due to interactions with dopamine [109]. Serotonin has also been linked to the default mode network (DMN), which is thought to be altered in ADHD [110,111]. Moreover, differences in TRP, KA, XA, and HAA serum levels in adult patients with ADHD compared to adult controls were reported [112]. This could confirm a connection between severity of ADHD symptoms and serum levels of tryptophan and tryptophan metabolites [112]. Furthermore, the positive correlations between the concentration of TRP and KYN in ADHD patients and control groups indicate a normal conversion of TRP to KYN. However, the number of studies focusing on the role of serotonin and tryptophan in aggression and ADHD is too limited and needs further research.
ijms-23-09968-t001_Table 1Table 1Summary of different biomarkers for detection/treatment of neuropsychiatric diseases.DisorderBiomarkerBiomarker ClassificationOutcome of FindingsRef.Depression (including MDD)CRP, IL-6, TNF-αInflammatoryThe simultaneous existence of three biomarkers in patients suffering from depression.[113]Alteration in δ/βactivity of EEGElectro-physiologicalSimultaneous decrement of δ-power and increase of β activity in the frontal lobe in people suffering from depression.[98]α1-anti-trypsinEpo-lipoprotein C3CortisolMyeloperoxidaseProteinThese biomarkers not only can be applied for MDD detection but also are able to estimate treatment response.[114]lncRNAsGeneticThe expression of lncRNAs can act as a promising biomarker for improving the detection of MDD in the clinical setting.[115]Angiotensin-converting enzymeBDNFCortisolProteomicPreparation of anecdotal evidence about the increment of pro-inflammatory/oxidative stress response in the acute stages of MDD.[116]Mood disorders (particularly anxiety)SB100ProteinElevation of SB100 in patients suffering from mood disorders, including anxiety.[117]SchizophreniaVariation in β and θ activityElectro-physiologicalThe surveys report the increase of β activity in all brain zones followed by increased θ activity in the upper temporal gyrus.[118]sTNF-R1, IL-6, IL-1Ra, OPG, vWf, sCD40L and hsCRP Pro-inflammatory markersInflammatoryThe study supports considerable negative associations between inflammatory markers and general cognitive abilities. [119]BDBDNFProteinSignificant decrease of BDNF in acute manic and BD.[120]ADHDSNAP-25 geneGeneticThe SNAP-25 T allele leads to a genetic load for ADHD.[121]PTSDBNPProteinThe amount of BNP is abnormally low in patients suffering from chronic PTSD.[122]Abbreviations: ADHD: attention deficit hyperactivity disorder; BD: bipolar disorder; BDNF: brain-derived neurotrophic factor; BNP: brain natriuretic peptide; CRP: C-reactive protein; hsCRP: high sensitivity CRP; IL: interleukin; MDD: major depressive disorder; PTSD: post-traumatic stress disorder; vWF: von Willebrand factor; SB100: S100 calcium binding protein B; TNF-α: tumor necrosis factor-α; OPG: osteoprotegerin.


## 6. Pharmacological Influences of TRP in Neuropsychiatric Diseases

Several investigations have been conducted to evaluate the feasibility of TRP application as an adjunctive to decrease the activity of 5-HT and MLT in disparate sorts of neuropsychiatric diseases [123,124]. The principle is based on the assumption that increased TRP accessibility for the brain will significantly improve the synthesis process of 5-HT [125]. Those biomedical protocols, which aim to reduce circulating TRP, are prevalent practices to evaluate the role of the 5-HT neurotransmission in different psychopathologies [11,57]. In a comprehensive review, Sarris et al. corroborated the encouraging impact of TRP adjunctive therapy with tricyclic antidepressants compared to placebo. These studies are older, dating back from the 1970s and 1980s, and no up-to-date research has been published, i.e., adjunctive TRP with newer antidepressants [126]. In a 4-week investigation on 12 uncontrolled aggressive schizophrenic patients, TRP supplementation significantly declined the number of events that needed intervention. All participating patients with schizophrenia with reported violent crimes were administered tryptophan and placebo in a double-blind crossover design. [127]. In another study, a double-blind placebo-controlled investigation with 98 healthy men and women was performed to assess the relationship between increased 5-HT and mood change (i.e., agreeable and quarrelsome manner). Astonishingly, it was found that TRP supplementation could considerably decrease aggressive/pugnacious behavior without any impact on mood or agreeableness [128]. The possible effect of TRP and thus 5-HT on social behavior was tested in 23 boys (age 10 years) with a history of increased physical aggression [129]. That was the first study to implement an acute TRP supplementation procedure with children. Findings showed participants in the TRP group tended to show more dominance, helpfulness, and affiliative responding. This pattern could be underlined by increased emotion regulation, leading to more situationally appropriate and goal-directed behaviors. Despite various investigations regarding the efficacy of TRP supplementation in neuropsychiatric disorders, challengeable arguments have remained among scientists due to some functional restrictions, including the low number of samples and poorly controlled processes [130]. 

Despite little application of TRP for chronic schizophrenia, this amino possesses excellent potential to improve mood and decrease hallucinations in patients suffering from schizoaffective disorders [130,131,132,133]. In a double-blind investigation, Brewerton and Reus concluded that a combination of lithium and TRP demonstrated superior efficiency compared to lithium plus placebo in manic schizoaffective patients [134]. Moreover, among 12 patients suffering from schizoaffective disorders, treatment with 8 gr of TRP for 4 weeks significantly decreased the symptoms of schizophrenia patients [135]. Since the decrement of 5-HT functions may contribute to the etiology of mania, TRP has been investigated as an antimanic agent [136]. Breaking down of serum TRP to kynurenine pathway can decrease the tryptophan required to prepare serotonin. Based on the monoamine-deficit assumption, a significant paucity of cerebral serotonin is included in chronic neuropsychiatric disorders such as mania and depression [15,61]. Due to the significant involvement of serotonin metabolite (melatonin) in disparate processes associated with BD (i.e., adjustment of circadian rhythms, sleeping, and immune cell reactivity), sensible shortage negatively impacts symptomatology and comorbidities [137]. In an experimental study, Murphy et al. corroborated that those 7 out of 10 patients who suffered from acutely manic or hypomanic disorders illustrated disease improvement when they received TRP (9.6 gr/day). The rest experienced relapse when a placebo was applied instead of TRP [138]. Pharmacological modification of the kynurenine pathway has been an interesting research subject in psychopharmacology due to its emphasis on cognitive performance, depression, and anxiety symptoms. The first investigation on TRP efficiency was published in 1963 by Coppen et al. [139]. In recent years, TRP demonstrated its great potential to improve the antidepressant influence of clomipramine, tricyclic antidepressants, and lithium [140,141,142,143]. It has been recently corroborated that focusing on the synthesis of the brain’s serotonin may increase the advancement of antidepressant drugs [144]. Biochemical facets of the serotonin deficiency in patients suffering from major depressive disorder are of great importance owing to the availability of the serotonin precursor tryptophan to the brain. This decrement is due to the precipitated degradation of tryptophan. It is worth noting that the induction of the extrahepatic IDO by the immune activation is unlikely to make a significant contribution [145]. Better speaking, the negligible antidepressant performance of TRP can be justified owing to its accelerated hepatic degradation. Improving TRP accessibility to the brain is vital to normalizing serotonin synthesis and may form the basis for future advancement of antidepressant drugs [144].

One of the essential advantages of TRP is improving sleep quality, followed by reducing the latency to sleep [146,147]. In another study, Schneider-Helmert and Spinweber presented a comprehensive overview to evaluate the influence of TRP on sleep and its efficiency in curing insomnia. The authors concluded that The TRP (from 1 to 5 g/day) possesses excellent potential to decline the latency to sleep as a hypnotic for chronic insomniacs without detrimental side effects such as decreased mental performance and intolerance [148]. In addition, TRP is an effective drug as a hypnotic for chronic insomnia at low doses. Interestingly, therapeutic TRP doses have reported no extreme side effects such as impaired psychomotor and mental performance [148]. This study has illustrated the efficiency of TRP in improving total sleep time and decreasing sleep fragmentation index and latency. Human experiments have implied that central 5-HT possesses the indisputable ability to modify an extensive array of cognitive processes in the current 20 years. Most of the evidence is obtained from psychopharmacological manipulations, which enhance or decline the activity of 5-HT in the brain. In humans, obtained information originates from investigations that have applied acute tryptophan depletion to enforce an acute global reduction in 5-HT synthesis in the brain [69,149,150]. However, there are serious concerns about the safety, side effects, and long-term use of psychostimulant medications [151]. TRP has been shown to increase serotonin levels in the brain and exhibit pharmacological effects. Moreover, tryptophan can also be used to synthesize other compounds, including niacin, proteins, and enzymes, which lead to numerous extra positive impacts on the user’s health.

It must be noted that although not all patients suffering from major depressive disorders might possess serotonin deficiency, the momentousness of TRP in the serotonin deficiency of major depressive disorders is indubitable [144,152,153,154,155]. Based on the abovementioned interpretation, liver TDO induction can describe the poor antidepressant performance of TRP. According to the reports, TRP is significantly effective in treating mild depression [156,157]. In current years, dietary patterns have been changed globally with a significant increment in the consumption of sugars, high-fat foods, and red meat. This substantial change has resulted in the emergence of gut microbiota alteration, which may contribute to the increased risk of chronic psychiatric disorders (i.e., depression) [158]. Population-based investigations have demonstrated the connection between traditional dietary patterns with risk of anxiety and depression [159]. It has been recently reported that the use of a ketogenic diet possesses great potential to modify the behavior in patients with psychiatric diseases, which implies the close connection between the gastrointestinal tract and schizophrenia [160]. Kim et al. reached the result that fermented foods can substantially improve cognitive function [161]. In addition, a gluten-free diet may be efficient in enhancing the levels of free L-TRP and modifying behavior, which is good evidence for the effect of leaky gut syndrome on gluten/casein sensitivity [162]. Several studies have illustrated the Mediterranean diet’s efficacy in reducing the risk of psychiatric disorders [163,164,165,166]. Generally, it is worth pointing out that decrement in the consumption of refined sugars and processed red meat and regular use of fermented foods and fish can efficiently avoid psychiatric disorders. Table 2 presents clinical research on targeting the gut–brain axis in neuropsychiatric diseases. Table 3 summarizes some clinical research outcomes on TRP therapeutic impacts in neuropsychiatric disorders.

## 7. Toxicology of TRP and Its Metabolites

Understanding the toxicity and pharmacological effects of TRP and its metabolites will be significantly helpful for future studies to use TRP as a therapeutic agent. However, very few data are reported on their toxic outcomes, and as such, further research is required. However, disruptions in tryptophan metabolism have been associated with various neurological and psychological disorders. Available data show that a TRP dose of 100 mg/kg body weight is usually well tolerated except for some slight gastric troubles [175]. In addition, some compounds in the KYN pathway are neurotoxic, and some are associated with CNS diseases. 3-hydroxkynurenine (3-HK) is one of the KYN pathway metabolites and its level in the brain is markedly elevated under several pathological conditions, including human immunodeficiency virus infection and hepatic encephalopathy. 3-HK causes neurotoxic effects due to hydrogen peroxide production, with consequent excessive hydroxyl radical and formation causing toxicity [176]. QA is an NMDA (N-methyl-D-aspartate) receptor agonist, causing increased neuronal firing, and leading to an axon-sparing neurodegenerative lesion, with lesion volume increasing proportionately to the QA concentration [177]. Previous studies showed a strong correlation between Huntington’s disease and the KYN pathway metabolites, including KYN, 3-HK, 3-HAA, QA, and KYNA. A study showed that for AIDS dementia complex patients, QA causes toxicity to cultured human astrocytes, which leads to glial cell and astrocyte loss [178]. Based on the impact of the KYN pathway on inflammation, Clark et al. previously suggested a link between the KYN pathway and cerebral malaria, possibly due to KYN pathway utilization in inflammatory responses mediated by microglia [179]. Higher levels of QA and PIC in cerebral malaria patients’ CSF were drastically associated with increased mortality rates [180]. Furthermore, several studies have noted indoxyl sulfate’s effects on uremic syndrome and its roles in cardiovascular disease, kidney and heart fibrosis, thrombogenicity, metabolic and hormonal dysfunction, inflammation, and chronic kidney disease mineral and bone disorder [181]. Indoxyl sulfate is responsible for many adverse effects and is a uremic toxin TRP metabolite. Type-2 diabetes is the primary cause of chronic kidney disease. Inflammation is associated with metabolic dysregulation in patients with type-2 diabetes and chronic kidney disease. Chronic kidney disease secondary to type-2 diabetes may be related to the accumulation of toxic TRP metabolites due to inflammation and impaired kidney function [182].

TRP is an essential part of the diet. However, its metabolism produces neurotoxic compounds. Therefore, further studies are required to understand the toxicological effects of these metabolites. Indeed, the TRP metabolites may be linked with several neurological diseases that warrant further investigation and can lead to opportunities for developing interventions to treat these conditions.

## 8. Conclusions and Future Perspectives

Neuropsychiatric diseases pose severe challenges that substantially deteriorate functioning in all aspects of life. These ailments possess a detrimental influence on psychosocial functioning. Due to the pleiotropic role of TRP in physiologic/pathophysiologic processes, its impact on human health is complicated. TRP is an important amino acid for the synthesis of proteins and energy metabolism. Due to the chemical activity of TRP metabolites and its multi-functionality at the systemic levels, TRP imbalance has been involved in disparate pathologic states such as CNS, autoimmune, and neuropsychiatric diseases. Participation in various physiologic functions has made this amino acid an essential disease biomarker with low specificity. Despite low specificity as a negative feature, TRP can be of great interest as a prognostic biomarker.

In this manuscript, a comprehensive study was attempted on the pharmacological effects of TRP in neuropsychiatric diseases. A presentation of the recent advancements in this paper was made to open new horizons towards a better perception of the role of TRP in neuropsychiatric diseases for those professionals involved in the field. For this aim, the role of TRP in various types of psychiatric disorders was discussed, and significant pathways toward its metabolism were presented. Eventually, TRP and its metabolites quantification following the neurotransmitter’s precursor were described. The identification of biomarkers with the appropriate capability of detecting/treating neuropsychiatric disorders is of great importance. Most neuropsychiatric diseases share symptoms and molecular pathways. Despite the significant influences of some biomarkers on neuropsychiatric disorders, their level of specificity can be controversial. In current years, numerous investigations have been carried out to highlight the ability of biomarkers for the detection/treatment of common neuropsychiatric disorders such as PTSD, MDD, BD, and schizophrenia. The most prominent parameter determining the potential application of biomarkers is their ability to alter various psychiatric and neurologic disorders, since commonly applied biomarkers are readily affected by environmental/lifestyle parameters like diet, stress, and substance abuse. Different types of inflammatory, protein, electrophysiological, genetic, and proteomic biomarkers possess great potential for application in diagnosing and treating neuropsychiatric disorders. Table 2 presents a comprehensive summary of applying the abovementioned biomarkers to detect/treat disparate neuropsychiatric disorders. As demonstrated, each biomarker has specific characteristics for opening new horizons to treat neuropsychiatric disorders. As a future perspective, further research is required towards the identification and role of biomarkers and their applications in therapeutic-associated purposes in neuropsychiatric disorders.

## Figures and Tables

**Table 2 ijms-23-09968-t002:** Recent studies targeting the gut–brain axis in neuropsychiatric disorders.

Disorder	Group of Study	Therapeutic Intervention	Therapeutic Outcome	Ref.
Schizophrenia	Patients who have Schizophrenia and Schizoaffective disorders	Ketogenic diet	Prevention of suicidal contemplation	[167,168]
			Improvement of concentration Reduction of hallucinations Improvement quality of life	[131]
Psychological stress	Students of medicine	*L.gasseri SBT2055 B.longum SBT2928*	Decreased stress	[169]
	Female volunteers	Probiotic *(L.casei Shirota)*	Enhancement of sleeping quality in stressful conditions	[170]
Anxiety and cognitive dysfunction	Male and Female volunteers	Probiotic *(L.casei Shirota)*	Improvement in quality of life	[171]
	Patients suffering from Fibromyalgia	ERGYPHILUS Plus *(L.Rhamnosus GG^®^, Casei, Acidophilus, and B. Bifidus*)	Improvement of decision-making capability	[172]
Mood disorders	Adults	Dietary fiber	Improvement of mood	[173]
Depression	Patients suffering from Major depression disorder	Probiotic *(L.Plantarum 299v*)	Improvement of cognitive function	[174]

**Table 3 ijms-23-09968-t003:** Clinical research outcomes of TRP therapeutic impacts on neuropsychiatric disorders.

Disorder	Clinical Outcome	Ref.
Schizophrenia	TRP supplementation significantly declined the number of events that needed intervention.	[127]
	TRP supplementation improves mood and decreases hallucinations in patients.	[131]
Mood disorders	TRP supplementation could considerably decrease aggressive/pugnacious behavior without any impact on mood or agreeableness.	[128]
	TRP supplementation enhances dominance, helpfulness, and affiliative responding.	[129]
Hypomanic disorders	Disease improvement after receiving TRP supplement.	[138]
Sleep disorders	TRP supplementation possesses excellent potential to decrease the latency to sleep as a hypnotic for chronic insomniacs.	[148]
Mild depression	TRP supplementation is significantly effective in treating mild depression.	[156]

## Data Availability

Not applicable.

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
