# Peer review of "The Role of Tryptophan Metabolites in Neuropsychiatric Disorders"

_ijms, 2022, doi:10.3390/ijms23179968_

Round 1

Reviewer 1 Report

The authors have submitted a review article of illustrating a current knowledge regarding impact of tryptophan metabolites such as serotonin, melatonin, and 5-OH-tryptophan on human neuropsychiatric disorders. The authors searched a range of eligible literature, from well-known classical, and latest research regarding an association of the tryptophan metabolites with depression, schizophrenia, ADHD, and so on, which are primarily attributed to the metabolic pathways. The authors discussed the beneficial availability of the tryptophan metabolites and the pharmacologic properties which ameliorate the states of the disease situation, resulting in reliable perspectives. This issue is of interest, and impact of their review is strong. My overall concern with the review describing the current available data regarding beneficial availability of the metabolic compounds listed in this review against various brain diseases is that information provided may offer something substantial that helps advance our understanding of effective management which draws novel class of effective compounds available in clinic. The reference list may be useful for readers who are interested in this issue.

Firstly, to strengthen authors’ perspectives, the authors are strongly recommended to add a “toxicology” sub-section regarding known tryptophan metabolites effect on humans, for instance. The opposite, toxicological effects of expected outcomes, if known, may influence largely the authors’ perspective.

Secondly, the authors should add a table demonstrating current knowledge for roles of tryptophan metabolites in treatment for neuropsychiatric disorders, instead of biomarkers. This is the point of their review.

Author Response

The authors appreciate your feedback. The authors have provided the required information in the manuscript. Line 368-405 and Table 3.

Reviewer 2 Report

Dear authors,

Congratulations for your important work. It is opening a new perception of the role of tryptophan in pathophysiology of several neuropsychiatric diseases as well as the potential possibilities for their treatment, what could be a topic for future research.

Author Response

The authors appreciate your feedback and comment.

Reviewer 3 Report

This review article entitled “The role of Tryptophan metabolites on neuropsychiatric disorders development” by Davidson et al. summarizes the impact of deficiency of tryptophan metabolism on various psychiatric diseases. Although the basic metabolic pathways and initial implication of its deficiency with psychiatric conditions are established ~50 years ago, studies are also performed in recent years on this topic, which this article covers well. There are several points that need clarification, including defining abbreviations as listed below:

L244

Describe what is exactly “aggressive” about this treatment.

L250

Describe the conclusion of this study (citation #120) would be informative to the reader.

L267

It’s unclear what is the “importance” and “decreased” compared to what.

L269

The sentence starting “Precipitated” is not clear.

L302

It sounds like 5-HT administration may have more direct effects than Trp. Advantages of treating with Trp, instead of 5-HT should be discussed here.

Define abbreviations when first appear such as:

L81      GI

L110    5-HT

L153    MLT

Author Response

Comment: “L244: Describe what is exactly “aggressive” about this treatment” The authors appreciate your feedback. The authors have provided the required information in the manuscript. Line 270-272. Comment: “L250: Describe the conclusion of this study (citation #120) would be informative to the reader” The authors appreciate your constructive comment. The authors have provided the required information in the manuscript. Line 278-282. Comment: “L267: It’s unclear what is the “importance” and “decreased” compared to what” The authors appreciate your constructive comment. We revised the sentence accordingly. Line 291-294. Comment: “L269: The sentence starting “Precipitated” is not clear” The authors appreciate your great feedback. We revised the sentence accordingly. Line 295-298. Comment: “L302: It sounds like 5-HT administration may have more direct effects than Trp. Advantages of treating with Trp, instead of 5-HT should be discussed here” The authors appreciate your constructive comment. The authors have provided the required information in the manuscript. Line 337- 341. Comment: “Define abbreviations when first appear such as: L81 GI L110 5-HT L153 MLT” The authors appreciate your constructive feedback. The authors revised the manuscript accordingly and all required information has been added to the manuscript.

Reviewer 4 Report

In this review, Majid Davidson and colleagues provide a detailed overview of the pharmacological role of tryptophan and its metabolites on the development of neuropsychiatric disorders. Overall, the review is well written and well structured. However, I have only some minor concerns and suggestions regarding the bibliography, which can be improved.

1.      The paragraph “Among KYN metabolites, quinolinic acid (QA) and kynurenic acid (KYNA)…shown great neuroprotection potential due to operating as an NMDA antagonist.” needs references (page 3, line 125-129)

2.      Please define UHPLC-ESI-MS/MS (page 3, line 147)

3.      In the sentence “Declined amount of TRP has been linked to different types of psychiatric disorders and clinical symptoms” (page 4, line 160) and in the sentences below, please specify whether in serum or in CSF.

4.      Ref 73 is used to define bipolar disorder (Bipolar disorder is one of the most critical chronic psychiatric disorders distinguished by recurrent depressive/manic episodes, page 4, line 179). However, it refers to the critical study of Fellendorf and colleagues, which I would suggest the authors to discuss better. Coherently, to provide a more appropriate reference for the bipolar disorder definition.

5.      In the discussion of the role of the tryptophane pathway in bipolar disorder, I suggest citing the comprehensive review doi: 10.3389/fimmu.2021.667179

6.      Regarding the role of lymphocytes T (page 5, line 203) in neuroinflammatory responses and in contributing to demyelination, I would suggest adding one more recent review (doi: 10.4103/1673-5374.198980).

7.      It would be interesting to briefly discuss the role of tryptophane in ADHD (the authors mention only one study in table 1 in this regard). I would suggest some important studies on the topic (doi: 10.1186/s12993-015-0080-x; doi: 10.1371/journal.pone.0032023; and the review doi: 10.1007/s00702-022-02478-5).

Author Response

1. The paragraph “Among KYN metabolites, quinolinic acid (QA) and kynurenic acid (KYNA)…shown great neuroprotection potential due to operating as an NMDA antagonist.” needs references (page 3, line 125-129) The authors appreciate your constructive comment. The authors have provided the required information in the manuscript. 2. Please define UHPLC-ESI-MS/MS (page 3, line 147) The authors appreciate your feedback. The authors have added the required information in the manuscript. Line 148. 3. In the sentence “Declined amount of TRP has been linked to different types of psychiatric disorders and clinical symptoms” (page 4, line 160) and in the sentences below, please specify whether in serum or in CSF. The authors appreciate your feedback. The authors have added the required information in the manuscript. Line 162-164. 4. Ref 73 is used to define bipolar disorder (Bipolar disorder is one of the most critical chronic psychiatric disorders distinguished by recurrent depressive/manic episodes, page 4, line 179). However, it refers to the critical study of Fellendorf and colleagues, which I would suggest the authors to discuss better. Coherently, to provide a more appropriate reference for the bipolar disorder definition. The authors appreciate your constructive feedback. The authors revised the manuscript accordingly and the required information has been added to the manuscript. Line 181-182. 5. In the discussion of the role of the tryptophane pathway in bipolar disorder, I suggest citing the comprehensive review doi: 10.3389/fimmu.2021.667179 The authors appreciate your constructive feedback. The authors revised the manuscript accordingly and the required information has been added to the manuscript. Line 192-195. 6. Regarding the role of lymphocytes T (page 5, line 203) in neuroinflammatory responses and in contributing to demyelination, I would suggest adding one more recent review (doi: 10.4103/1673-5374.198980). The authors appreciate your constructive feedback. The authors revised the manuscript accordingly and the required information has been added to the manuscript. 7. It would be interesting to briefly discuss the role of tryptophane in ADHD (the authors mention only one study in table 1 in this regard). I would suggest some important studies on the topic (doi: 10.1186/s12993-015-0080-x; doi: 10.1371/journal.pone.0032023; and the review doi: 10.1007/s00702-022-02478-5). The authors appreciate your feedback. The authors have added the required information in the manuscript. Line 236-250.

Round 2

Reviewer 1 Report

The authors have addressed properly all the issues raised by reviewers including me. I have no more comments, and now recommend that this manuscript is acceptable for publication in the journal IJMS.